

# Pseudotargeted lipidomics analysis of scoparone on glycerophospholipid metabolism in non-alcoholic steatohepatitis mice by LC-MRM-MS

Qi Song[1,2], Ziyi Zhao[1], Hu Liu[1], Jinling Zhang[1], Zhiqiang Wang[3], Yunqi Zhang[1], Guowei Ma[1] and Shaoqin Ge[1,4]

[1] College of Traditional Chinese Medicine, Hebei University, Baoding, Hebei, China
[2] Institute of Chinese Materia Medica, China Academy of Chinese Medical Sciences, Beijing, China
[3] Hebei Key Laboratory of Public Health Safety, School of Public HealthPublic Health, Hebei University, Baoding, Hebei, China
[4] College of Basic Medical Science, Hebei University of Technology, Baoding, Hebei, China

Corresponding author
Shaoqin Ge, gesq67@163.com

## ABSTRACT

As the inflammatory subtype of nonalcoholic fatty liver disease (NAFLD), the progression of nonalcoholic steatohepatitis (NASH) is associated with disorders of glycerophospholipid metabolism. Scoparone is the major bioactive component in *Artemisia capillaris* which has been widely used to treat NASH in traditional Chinese medicine. However, the underlying mechanisms of scoparone against NASH are not yet fully understood, which hinders the development of effective therapeutic agents for NASH. Given the crucial role of glycerophospholipid metabolism in NASH progression, this study aimed to characterize the differential expression of glycerophospholipids that is responsible for scoparone's pharmacological effects and assess its efficacy against NASH. Liquid chromatography-multiple reaction monitoring-mass spectrometry (LC-MRM-MS) was performed to get the concentrations of glycerophospholipids, clarify mechanisms of disease, and highlight insights into drug discovery. Additionally, pathologic findings also presented consistent changes in high-fat diet-induced NASH model, and after scoparone treatment, both the levels of glycerophospholipids and histopathology were similar to normal levels, indicating a beneficial effect during the observation time. Altogether, these results refined the insights on the mechanisms of scoparone against NASH and suggested a route to relieve NASH with glycerophospholipid metabolism. In addition, the current work demonstrated that a pseudotargeted lipidomic platform provided a novel insight into the potential mechanism of scoparone action.

## INTRODUCTION

Nonalcoholic steatohepatitis (NASH) is caused by the interaction of several potential factors, including lipotoxicity, mitochondrial dysfunction, oxidative stress and liver endoplasmic reticulum (ER) stress, showing the characteristics of fat accumulation, liver inflammation and hepatocyte ballooning, which can evolve into liver fibrosis, cirrhosis and
hepatocellular carcinoma (*Sheka et al., 2020*). With the continuously rising global burden of obesity, NASH is currently one of the most frequent causes of liver transplantation and is predicted to become an even larger problem in coming decades (*Younossi et al., 2019*). Several therapeutic strategies that target various stages in the development of steatosis and steatohepatitis have been evaluated, yet only limited therapies are currently available (*Ratziu, Francque & Sanyal, 2022*). Due to the growing incidence of NASH, it remains urgent to develop a suitable pharmacotherapy.

Glycerophospholipids, as the most abundant and ubiquitous phospholipids in mammalian cellular membranes, are a critical series of compounds of membrane structures forming cellular organelles and cell–cell barriers and are involved in biological process regulations, such as inflammatory stress, lipotoxicity, and lipid signaling (*Mukhopadhyay & Trauner, 2023*). Dysregulation of glycerophospholipids in liver is suggested as a potential key link in the mechanism of disease progression toward NASH in humans (*Gorden et al., 2015*). Hepatocytes have a high demand for glycerophospholipids, which play a key role in promoting the assembly and secretion of very-low-density lipoprotein triglyceride (*Fu et al., 2011*; *Kim et al., 2018*; *Van der Veen et al., 2017*; *Walker et al., 2011*). The metabolic disorder of glycerophospholipids is related to the destruction of hepatocyte membrane functional integrity, resulting in the release of extracellular lipotoxic lipids (*Payne et al., 2014*). Glycerophospholipids on the lipid droplet surface can promote multiple lipid droplets to aggregate into bigger ones (*Hafez & Cullis, 2001*). In addition, glycerophospholipids have been related to multiple deleterious effects on the liver, such as increased cholesterol biosynthesis, reductions in mitochondrial fatty acid oxidation, increased mitochondrial permeability and swelling (*Paul, Lewinska & Andersen, 2022*), as well as ER stress and hepatocyte apoptosis (*Liu et al., 2020b*).

As one of the major bioactive components in *Artemisia capillaris*, scoparone has been shown to be efficacious in treating liver disorders, shows hepatoprotectivity, and contributes directly to the therapeutic effect (*Gao et al., 2020*; *Hui et al., 2020*; *Liu & Zhao, 2017*; *Wang et al., 2016*). The antihyperlipidemic properties of scoparone and other coumarin derivatives were investigated in a rat model of $CCl_4$-induced hepatic injury-dependent hyperlipidemia (*Taşdemir et al., 2017*). The results indicated that scoparone pretreatment protected from $CCl_4$-induced elevation in serum total cholesterol, triglyceride, low-density lipoprotein levels and a concomitant decrease in high-density lipoprotein levels. Subsequently, studies have found that scoparone could protect the liver against alcohol-induced liver injury by regulating the levels of PE(19:1(9Z)/0:0), PE(17:1(9Z)/0:0), PG(19:1(9Z)/14:0) using a lipidomics strategy (*Zhang et al., 2016*). However, there is limited information to evaluate the alterations in glycerophospholipid metabolism of the hepatoprotective effects of scoparone against NASH. Herein we employed a pseudotargeted lipidomic approach to analyze glycerophospholipids by using ultra-performance liquid chromatography (UPLC)-hybrid triple quadrupole-linear ion trap (QTRAP)-MS/MS in the time-scheduled multiple reaction monitoring (MRM) mode. The present study might provide detailed new insights into the role of scoparone in the regulation of glycerophospholipid metabolism that is of value in understanding the potential treatment of NASH.

## MATERIALS & METHODS

### Drug and chemicals

The 42 kinds of glycerophospholipids isotope-labeled internal mix standards were obtained at Avanti (Birmingham, AL, USA) (Table S1). All reagents were HPLC grade and used for sample preparative and analytical UPLC analysis (Thermo Fisher Scientific, Waltham, MA, USA). Scoparone (≥98%, HPLC, CAS number 120-08-1) was supplied by purechem-standard Corp. (Chengdu, China). High-fat diet (HFD: 45% energy from fat) and normal chow diet (NCD: 15% energy from fat) were purchased from Keao Xieli (Beijing, China).

### Experimental animals and design

The experiments were approved by the animal management regulations of the Ethical and Animal Welfare Committee of Hebei University (No. IACUC-2019016XG). 4-week-old male C57BL/6 mice, weighing 18–22 g, were purchased from Qinda (Qingdao, China). Animals were acclimatized to their housing environment for 1 week prior to experimentation. The temperature of the housing facility was maintained at $24 \pm 1\ °C$ with a relative humidity of $60 \pm 5\%$ and a 12-h day-night cycle. They were subsequently randomized into three groups, with 8 mice per group in the control group, NASH group and scoparone group. After 1 week of acclimation, the control group was fed the NCD while other groups were fed the HFD for 20 weeks. At week 21, olive oil (control and NASH groups) or scoparone (scoparone group) was administered by gavage (50 mg/kg) for seven consecutive days.

After the end of 7-day treatment, all mice were humanely sacrificed. Liver samples were collected, and blood was immediately centrifuged (579 g for 15 min at 4 °C). The liver tissue samples were stored at −80 °C for subsequent lipidomic analysis.

### Biochemical assays

The serum levels ($n = 8$ per group) of triglyceride (TG), aspartate aminotransferase (AST) and alanine aminotransferase (ALT) were detected by commercially available kits (Millipore, Burlington, MA, USA), according to the manufacturer's instructions. These assay kits were purchased from Jiancheng Bioengineering (Nanjing, China).

### Hematoxylin and eosin and oil red O staining analysis

For hematoxylin and eosin (H&E), liver tissues were fixed within 4% paraformaldehyde for 24 h, followed by dehydration, paraffin embedding, serially sectioning (4μm), and staining with H&E for evaluation of histopathological changes. Liver sections for Oil Red O staining were cut 8 μm-thick, fixed with 4% paraformaldehyde and stained with hematoxylin and Oil Red O. These reagents were obtained from Solarbio Life Science (Beijing, China).

### Lipid extraction and sample preparation

Hepatic lipids ($n = 8$ per group) were extracted from 30 mg of liver tissue by homogenizing in 300 μL methanol: water solution (v/v =1:1) (*Song et al., 2022*). Each homogenate was added to 300μl chloroform, vortexed and then sonicated for 10 min. After the lower layer (chloroform fraction) was taken by centrifugation, 300 μL chloroform/methanol mixture (v/v =2:1) was added for extraction followed by vortexing and then sonicating for 10

min. After centrifugation and recovery of the lower layer of chloroform, both chloroform extracts were combined and evaporated completely. The lipid residue was dissolved in 200 μL isopropanol-methanol solution (v/v =1:1), and 20 μL mixed isotope internal standard was added. After vortex and centrifugation, the supernatant was collected for LC-MRM-MS analysis.

## Quantitative phospholipid profiling by LC-MRM-MS

**Chromatography.** The LC system was carried out using an ExionLC System that consisted of a binary high-pressure mixing gradient pump with a degasser, a column oven, and a thermostated autosampler. Finally, the conditions were optimized as follows:

The injection volume was 5 μL, and the autosampler temperature was held at 10 °C. Eluents used were 4:6 water/acetonitrile (eluent A, v/v) and 9:1 methanol/acetonitrile (eluent B, v/v), both added with 10 mM ammonium formate and 0.1% formic acid. The column temperature was 55 °C, and the flow rate was 0.35 mL/min. Chromatographic separation was performed using a UPLC HSS T3 (1.7 um, 2.1 × 100 mm, Waters). The elution gradient began at 0% B for 1.5 min, then a linear increase to 55% at 5 min, 60% at 10min, 70% at 13 min, 90% at 15 min and 100% B at 16 min, and then held at 100% for 2 min. Thereafter, the system returned to the initial conditions and was held constant for up to 2 min to allow column equilibration.

**Mass spectrometry.** The MS was QTRAP® 6500+ (Sciex, Framingham, MA, USA) fitted with an IonDrive™ Turbo V source. MS data were performed using the scheduled multiple-reaction monitoring (MRM) mode operating in the positive- and negative-ion ESI mode. Source condition was as follows: ion spray voltage was −4.5KV/+5.5KV, collision gas was medium, curtain gas was 40psi, ion source temperature was 400 °C, gas 1 was 50 psi, and gas 2 was 55 psi.

## Data processing

All data for samples were uploaded to MRMPROBS software (RIKEN Center for Sustainable Resource Science, Saitama, Japan) for automated data acquisition, processing, and report generation. Integrated peak area of metabolites was used to calculate the relative concentration of each detected lipid. The formula is as follows: $CA = A1 / A2 * C * V / N$; where, CA, analyte concentration (ng/mL); A1, analyte intensity; A2, internal standard intensity; C, internal standard concentration (mg/mL); V, constant volume (0.2mL); N, sample volume. The data were output to Microsoft Excel and the concentrations were reported as ng/mL in the reconstituted solution. The multivariate data analysis was performed using Ezinfo3.0 software (Waters Corp., Milford, MA, USA) for the unsupervised principal components analysis (PCA) and orthogonal projection to latent structure-discriminant analysis (OPLS-DA). OPLS-DA was utilized to validate the PCA model. The furthest metabolites from the origin showing a higher value of VIP-plots generated from OPLS-DA were potential biomarkers.

## Statistical analysis

All statistical analyses were performed with SPSS 19.0 statistical analysis software (IBM, Armonk, NY, USA). A value of $p < 0.05$ was considered statistically significant.

## RESULTS

### Scoparone alleviates HFD-induced hepatic steatosis and injury

To examine the effects of scoparone on NASH, the mice were fed with an HFD for up to 20 weeks. As expected, HFD-fed animals accelerated weight gain and increased serum TG, AST, and ALT levels, reflecting hepatocyte damage. With the daily bodyweight of the mice measured over the course of the scoparone treatment, a significant weight change was observed in the 5-day treatment ($p < 0.05$) (Fig. 1A). After 7 days of treatment with scoparone, the levels of serum TG, AST and ALT had been dramatically reduced ($p < 0.01$) (Figs. 1B, 1C and 1D).

H&E staining of liver tissues (Fig. 1E) exhibited numerous large lipid droplets (hepatic steatosis), local infiltration of inflammatory cells (hepatic inflammation) and diffuse vacuolization of hepatocellular cytoplasm (hepatocellular ballooning) in the NASH group compared with the control group. Strikingly, scoparone treatment effectively improved hepatocellular ballooning, hepatic steatosis, and inflammation in HFD-fed mice. The above histological outcomes indicating the effects of scoparone on hepatic steatosis were confirmed by staining with Oil Red O (Fig. 1F). Results showed that hepatic fat accumulation was apparent in NASH mice, while scoparone strongly suppressed the effect of HFD on hepatic steatosis. Collectively, these results suggested that scoparone robustly inhibited hepatic inflammation, hepatic steatosis, and hepatocellular ballooning.

### The glycerophospholipid profile analysis by LC-MRM-MS

The representative LC-MRM-MS total ion flow chromatogram of each group showed good peak shape, separation, and intensity, indicating that the chromatographic and MS conditions optimized were suitable for the liver analysis in this study (Fig. 2). A total of 340 glycerophospholipids were detected with quantitative thresholds in liver. The stability and reproducibility of the data were assessed by the quality control (QC) samples during the experiment (Fig. S1). After peak alignment, filtering and normalization, the RSD of intensity of the above glycerophospholipids in repeated QC sample measurement was less than 40%. Unsupervised PCA analysis was employed to visualize the separation between the control and NASH group (Figs. 3A and 3B). Obvious separation was observed in the OPLS-DA analysis, indicating the unique differential glycerophospholipids expression of each group (Figs. 3C and 3D). In order to discover potentially valuable disease-related biomarkers, glycerophospholipids that differed with statistical significance between the control and NASH groups after correcting for multiple comparisons were screened. Variables that made significant contributions to the classification were determined by a threshold of VIP values, which could be obtained from OPLS-DA (Figs. 3E and 3F). A total of 14 glycerophospholipids associated with HFD-induced mouse model were screened following the above-mentioned criteria and performed clustering heat map analysis (Fig. 4). In addition, the levels of these glycerophospholipids were significantly changed in the NASH group compared with the control group (Fig. S2).

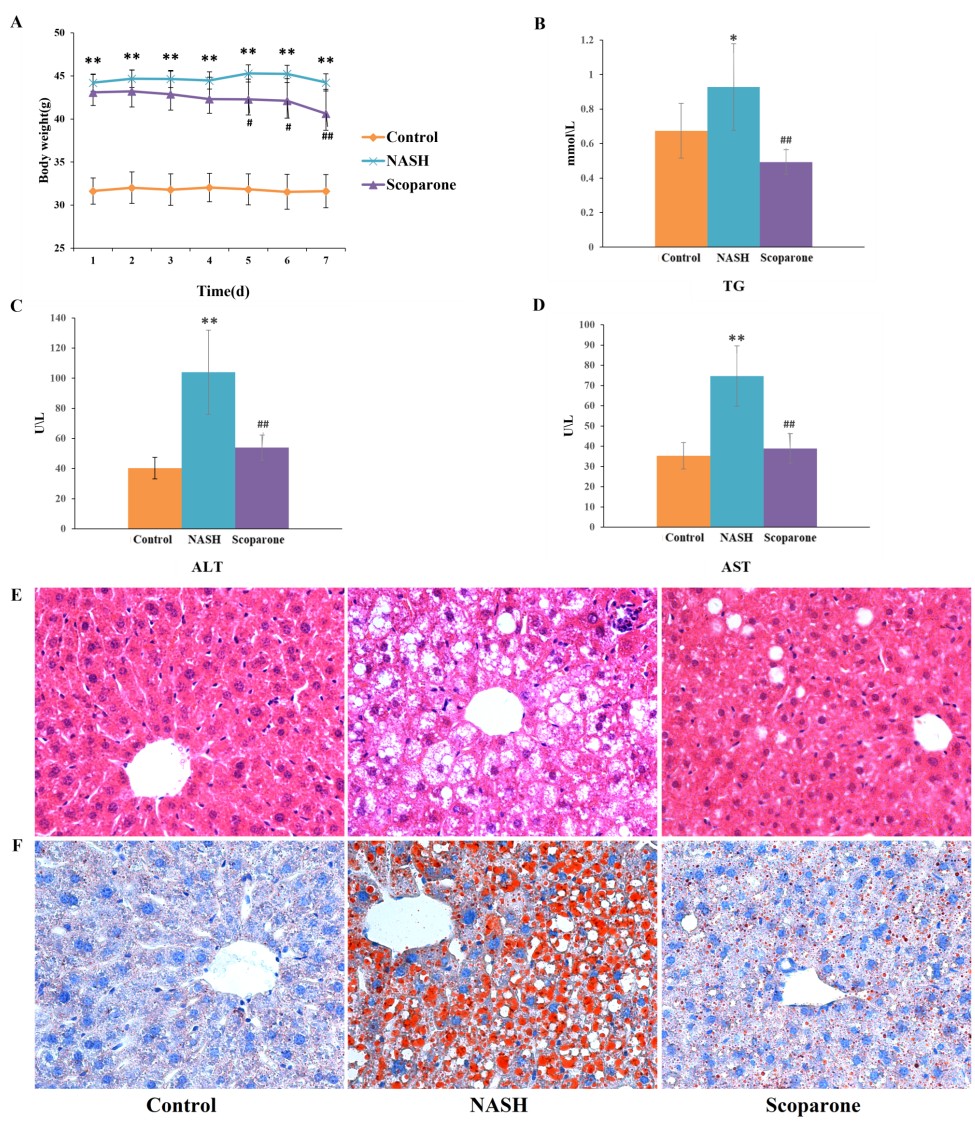

**Figure 1  Scoparone diminishes steatosis and hepatic inflammation in HFD-fed NASH-like mice.** (A) Mice weight 7 days after drug administration; (B) Serum TG; (C) Serum ALT; (D) Serum AST; (E) HE-stained liver tissue, magnification, ×400; (F) Oil red O-stained liver tissue, magnification, ×400. Data are mean ± SD, $n = 8$. *$P < 0.05$ and **$P < 0.01$ compared with the control group; #$P < 0.05$ and ##$P < 0.01$ compared with the NASH group.

## Effect of NASH treated with scoparone based on metabolic profile

The PCA analysis of liver samples revealed distinct separation among the control, NASH and scoparone groups (Figs. 5A and 5B). The OPLS-DA score plot revealed the metabolic trajectory of the scoparone group was similar to the control group and in the direction away from the NASH group (Figs. 5C and 5D). The apparent trend suggested that the pathological process of NASH might be reversed by scoparone.

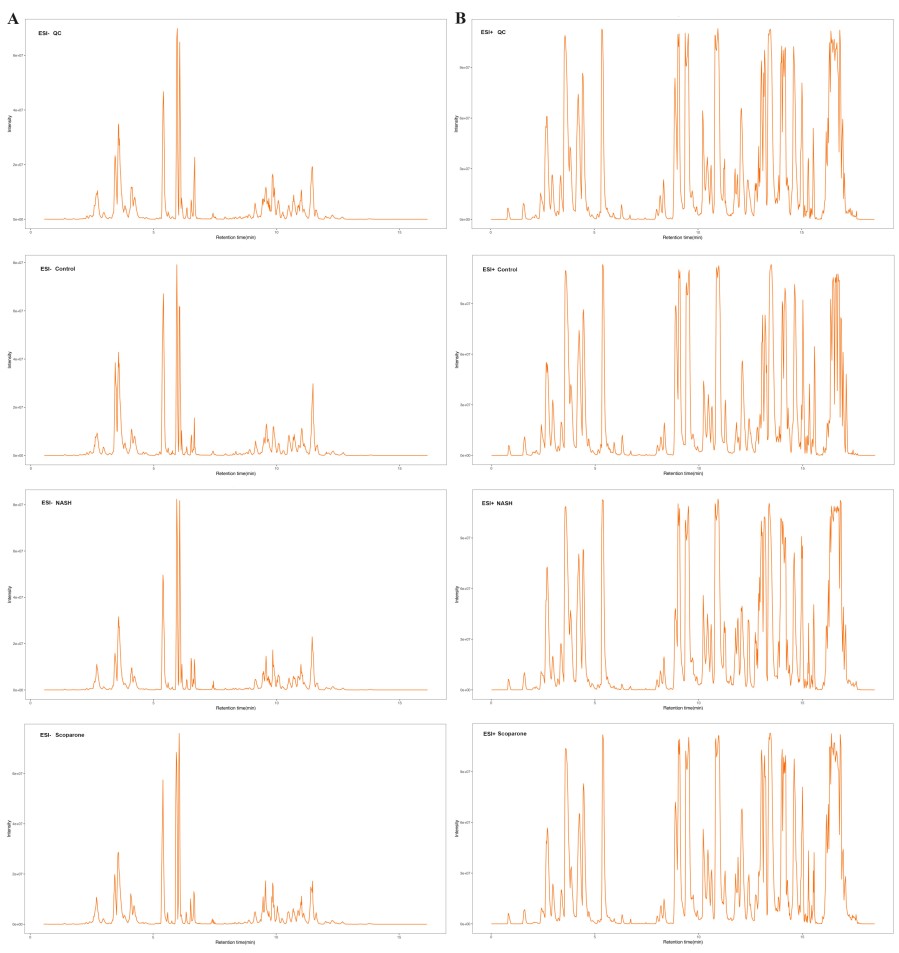

**Figure 2** The typical total ion chromatograms of QC, control, NASH and scoparone samples in the negative mode (A) and the positive mode (B).

## Scoparone ameliorates metabolic dysregulation of glycerophospholipids in NASH mice

Lipidomic analysis showed that the amount of phosphatidylethanolamine (PE) (18:1/20:4), lysophosphatidylcholine (LPC) (18:0), phosphatidylcholine (PC) (16:0/20:4), PE(16:0/20:4), PE(18:1/20:3), PE(18:0/22:4), PC(18:1/20:4), and PE(18:0/22:5) was higher in the NASH group compared with the control group (Fig. 6A). In parallel, HFD-fed mice had a notable decrease of PC(18:0/18:2), PC(16:0/20:5), and PC(18:2/20:2). It was noteworthy that these glycerophospholipids exhibited clear differences in mice after taking scoparone. Scoparone had striking and consistent increase of PC(18:0/18:2), PC(16:0/20:5), and PC(18:2/20:2) compared to HFD-fed mice, significantly reduced PE(18:1/20:4), LPC(18:0), PE(16:0/20:4), PC(16:0/20:4), PE(18:1/20:3), PE(18:0/22:4), PC(18:1/20:4), and PE(18:0/22:5) as anticipated. Correlation analyses showed significant relationships between the above glycerophospholipids (Fig. 6B). In summary, HFD-fed

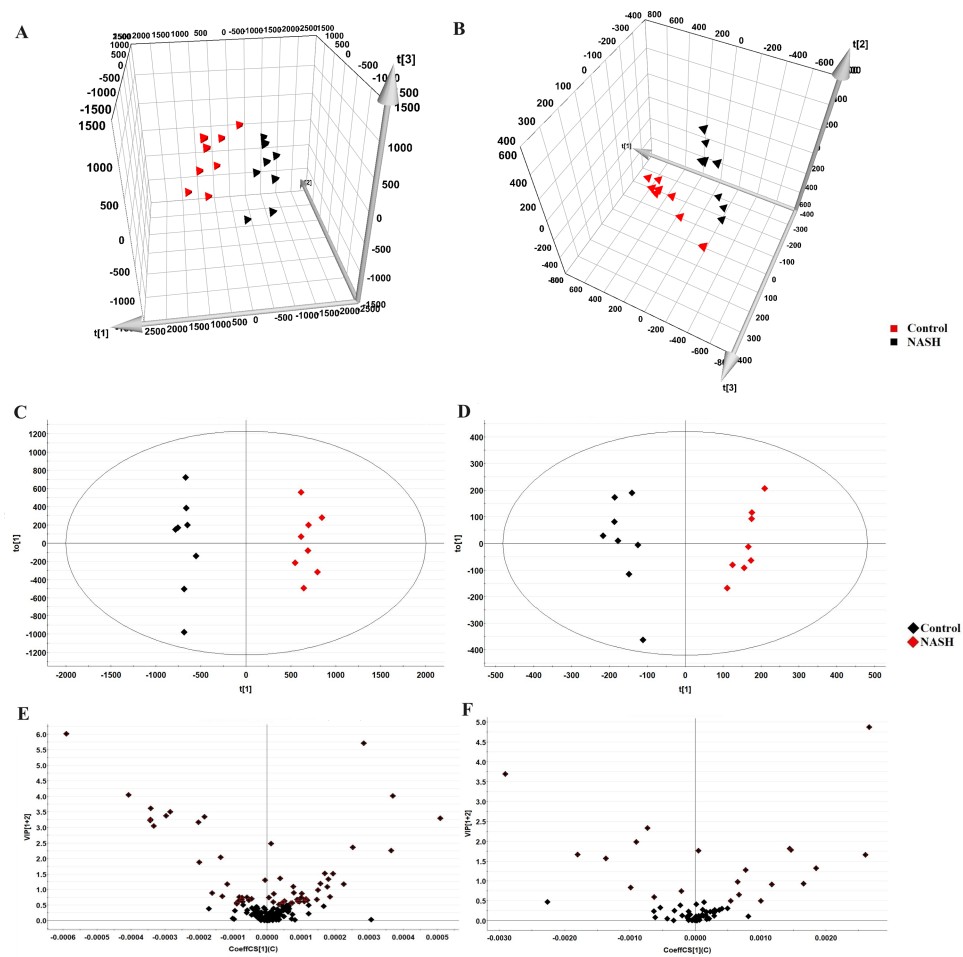

**Figure 3 Multivariate data analysis of liver samples from the control and NASH group.** 3D PCA score plot in the negative mode (A) and positive mode (B); OPLS-DA plot in the negative mode (C) and positive mode (D); VIP plot of OPLS-DA model in the negative mode (E) and positive mode (F).

induced dysregulations of glycerophospholipids, which were found to be regulated with scoparone treatment.

## DISCUSSION

Recent epidemiological data suggest that the prevalence of NASH among the general adult population is 3–5%, being the second most common indication for liver transplant (*Sayiner et al., 2018*). However, there are no therapies approved for NASH, and non-pharmacologic treatment aimed at reducing fatty liver through exercise and body weight loss is recommended (*Vuppalanchi et al., 2021*). Therefore, it is urgent to seek for the ideal drugs that can alleviate the progress of NASH.

Scoparone is the main active ingredient of *Artemisia capillaris*, an effective Chinese herbal medicine for NASH. Although the mechanism of action has not been fully elucidated, the therapeutic effect of scoparone is to suppress inflammation and further improve NASH by

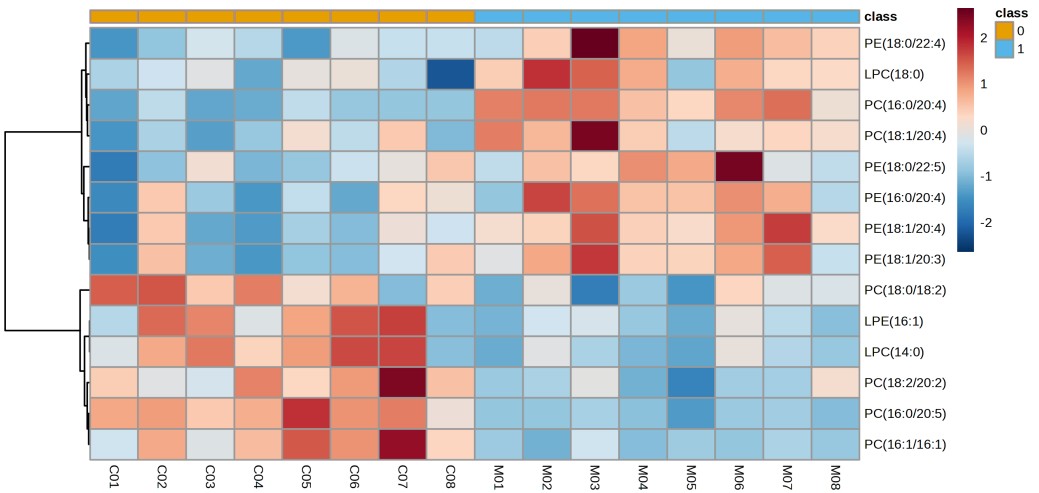

**Figure 4** Heatmap visualization of glycerophospholipids linked to HFD-fed NASH-like mice.

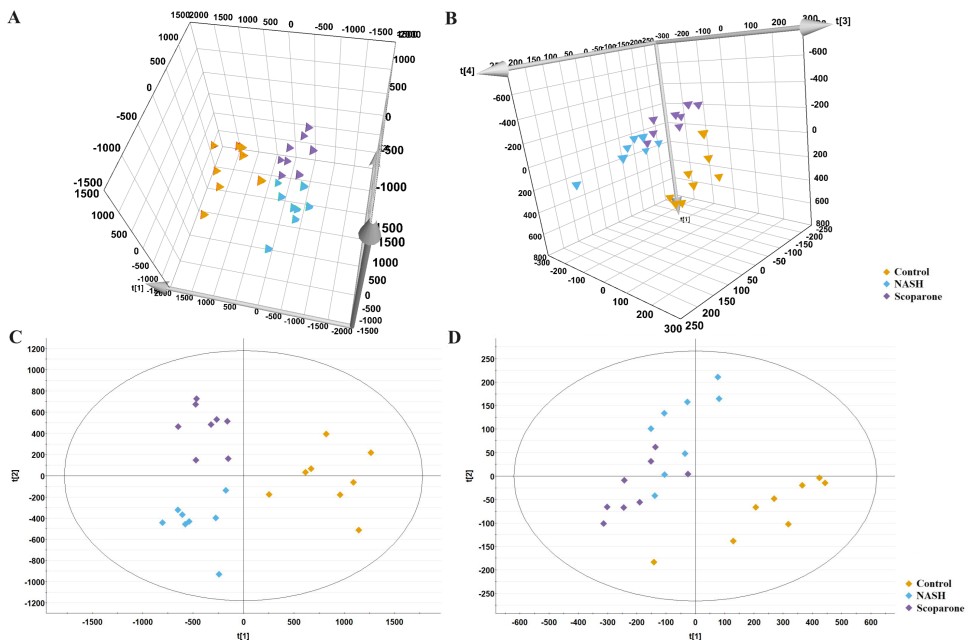

**Figure 5** Metabolic profiles of liver samples in mice with NASH induced by scoparone. PCA score plot in the negative mode (A) and positive mode (B); OPLS-DA plot in the negative mode (C) and positive mode (D).

inhibiting the PI3K/AKT/mTOR, ROS/P38/Nrf2 axis, and TLR4/NF-$\kappa$B signaling pathway (*Liu et al., 2019*; *Liu et al., 2020a*). Interestingly, it has been reported that scoparone also reduces the accumulation of fat *via* suppression of PPAR $\gamma$ to repress adipogenic gene expression in 3T3-L1 preadipocytes (*Noh et al., 2013*). Therefore, it is likely that scoparone attenuates NASH by acting on multiple biological pathways, yet lipid mechanisms of

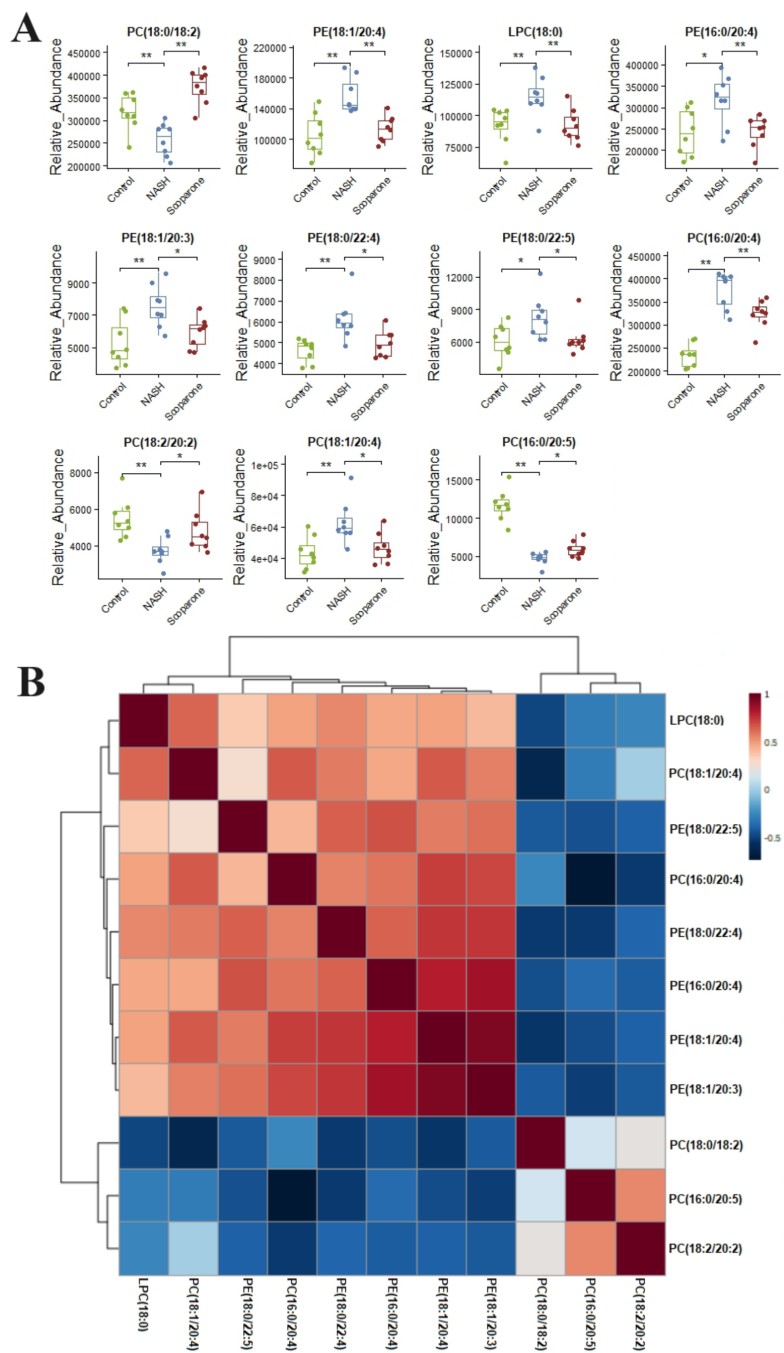

**Figure 6 Histogram visualization for the treatment of scoparone on NASH.** Bar chart visualization of the phenotypic characterization of NASH (A); correlation analysis of the eleven endogenous metabolites using the metaboAnalyst 5.0 software (B). The single asterisk (*) indicates $p < 0.05$, while the double asterisk (**) indicates $p < 0.01$.

scoparone against NASH remain incompletely understood. In order to shed some light on the levels of glycerophospholipids of scoparone against HFD-induced mice, we conducted lipidomic analysis in a NASH animal model.

With the ideal analytic stability and fast scan speed, LC-MRM-MS have proven to be powerful tools for lipidome studies (*Chen & Zhang, 2020*; *Jung et al., 2020*). Examining the specific lipid classes to evaluate candidate medicines, we first performed a deeper study of scoparone on HFD-induced glycerophospholipids changes in mice *via* pseudotargeted lipidomics using LC-MRM-MS. In total, 14 altered glycerophospholipids associated with HFD-induced mouse model were identified in liver. Interestingly, the levels of 11 glycerophospholipids were found to be modulated in response to scoparone treatment. Glycerophospholipids changed significantly under scoparone intervention, including PC(18:0/18:2), PC(16:0/20:5), PC(18:2/20:2), LPC(18:0), PE(18:1/20:4), PE(16:0/20:4), PC(16:0/20:4), PE(18:1/20:3), PE(18:0/22:4), PC(18:1/20:4), and PE(18:0/22:5). The levels of identified PE(18:1/20:4), LPC(18:0), PE(16:0/20:4), PC(16:0/20:4), PE(18:1/20:3), PE(18:0/22:4), PC(18:1/20:4), and PE(18:0/22:5) that have been shown up-regulated in the NASH group compared with the control group were found to be down-regulated in the scoparone group compared with the NASH group. The levels of identified PC(18:0/18:2), PC(16:0/20:5), and PC(18:2/20:2) that have been shown down-regulated in the NASH group compared with the control group were found to be up-regulated in the scoparone group compared with the NASH group.

In obese individuals, adipocytes turn into hypertrophy, and the metabolism of PE and PC is dysregulated. On the other hand, the dysregulations of PE and PC may cause hepatotoxicity and induce NASH *via* various mechanisms such as lipid droplet accumulation (*Arendt et al., 2013*). In patients with NASH, the levels of PE and PC were correlated with disease severity (*Alcoriza-Balaguer et al., 2019*). The alterations in PE and PC metabolism in liver with steatosis or steatohepatitis could be observed (*Li et al., 2006*). Previous studies reported that PE and PC could be a common response to hepatocyte stress, inflammation, and lipid accumulation (*Elblehi, Hafez & El-Sayed, 2019*; *Guo & Davies, 2013*; *Kitagawa et al., 2015*).

LPC is a potential bioactive lipid found in hepatocytes that is involved in the induction of inflammatory responses (*Trovato et al., 2021*). Both exogenous and endogenous LPC can induce hepatocyte lipoapoptosis, which is related to endoplasmic reticulum stress (*Chen et al., 2022*). As patients with NASH show increased hepatic LPC content, this dysregulation is likely to contribute further to the disease's progression (*Liangpunsakul & Chalasani, 2018*). Instead, a lower intracellular LPC concentration could attenuate fatty acid uptake of hepatocytes, thereby reversing the experimental NASH (*Kawano et al., 2015*).

In summary, our experiments indicate that scoparone attenuates HFD-induced lipid accumulation and hepatic inflammation, regulating glycerophospholipid metabolism. Overall, our data shows for the first time how scoparone regulates dysregulations of glycerophospholipid metabolism induced by HFD-fed, and has a rudimentary understanding of the corresponding mechanism.

## CONCLUSIONS

As far as we know, this is the first study showing that scoparone alleviates NASH *via* specifically targeting glycerophospholipids based on a pseudotargeted lipidomic strategy. All in all, in this study, we reported that scoparone regulated glycerophospholipid metabolism during the pathogenesis of NASH and also acted as an important anti-lipotoxic and anti-inflammatory. The positive results confirm and extend the understanding of the regulatory mechanism of scoparone on NASH, representing a potential avenue for the amelioration of NASH.

### Funding

This work was supported by grants from the Natural Science Foundation of Hebei Province (No. H2019201426), the Science and Technology Project of Hebei Education Department (No. QN2021017), Medical Science Foundation of Hebei University (No. 2021A08), College Students' Innovative Entrepreneurial Training Plan Program (No. 2022379). The funders had no role in study design, data collection and analysis, decision to publish, or preparation of the manuscript.

### Grant Disclosures

The following grant information was disclosed by the authors:
Natural Science Foundation of Hebei Province: H2019201426.
Science and Technology Project of Hebei Education Department: QN2021017.
Medical Science Foundation of Hebei University: 2021A08.
College Students' Innovative Entrepreneurial Training Plan Program: 2022379.

### Competing Interests

The authors declare there are no competing interests.

### Author Contributions

- Qi Song conceived and designed the experiments, performed the experiments, analyzed the data, prepared figures and/or tables, and approved the final draft.
- Ziyi Zhao performed the experiments, prepared figures and/or tables, and approved the final draft.
- Hu Liu performed the experiments, prepared figures and/or tables, and approved the final draft.
- Jinling Zhang performed the experiments, prepared figures and/or tables, and approved the final draft.
- Zhiqiang Wang conceived and designed the experiments, authored or reviewed drafts of the article, and approved the final draft.
- Yunqi Zhang performed the experiments, prepared figures and/or tables, and approved the final draft.
- Guowei Ma performed the experiments, prepared figures and/or tables, and approved the final draft.

- Shaoqin Ge conceived and designed the experiments, authored or reviewed drafts of the article, and approved the final draft.

### Animal Ethics

The following information was supplied relating to ethical approvals (*i.e.*, approving body and any reference numbers):

Hebei University Affidavit of Approval of Animal Welfare and Ethical

### Data Availability

The raw data are available in the Supplemental Files.

### Supplemental Information

Supplemental information for this article can be found online at http://dx.doi.org/10.7717/peerj.17380#supplemental-information.

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
