# Peer review of "Pseudotargeted lipidomics analysis of scoparone on glycerophospholipid metabolism in non-alcoholic steatohepatitis mice by LC-MRM-MS"

_PeerJ, doi:10.7717/peerj.17380_

## Round 0.1 · original submission · Major Revisions

One of the reviewers has made a significant number of suggestions for improvement which you should carefully and completely address.

However, I would also add that the following must be fully answered as I am concerned about the statistical analysis and robustness of this work, and its rationale.

First, provide a power calculation to show that your (small) animal group sizes are sufficiently powered to detect changes.

All statistical analyses require you to explain how many biological and technical replicates were performed for every experiment presented. This information is essential.

Finally, why is this study of merit? As noted by reviewer 2, the therapeutic benefits of this are already clear, so what advance is being reported here? It is essential that this is clearly stated and a solid reason given, as without this there is little merit to the study.

Please address these points fully and clearly in any revision you may wish to resubmit for further review.

Reviewer 1 ·

Basic reporting

The article is well written, understandable and easy to follow. Hypothesis is well defined with sufficient background.

Experimental design

Research is well designed, following ethical standards. Methodology is described in details and well explained. Author could add that body weight was measured and how often? Why did authors choose 7 days-treatment?

Validity of the findings

results are well presented, graphs are clear and well explained.

Additional comments

In introduction, authors mentioned NASH as a leading cause for liver transplantation. However, even more common are liver failure, cirrhosis and hepatocellular carcinoma.

Reviewer 2 ·

Basic reporting

The study described the changes in glycerophospholipid metabolism in high-fat diet-induced non-alcoholic steatohepatitis mice compared to controls and demonstrated the abilities of scoparone, a bioactive component of Yinchenhao, in reversing the trends. Overall, the use of LC-MRM-MS provided an appropriate scientific tool/method for the glycerophospholipid study. I also appreciate that the authors carried out orthogonal biochemical and histopathological assays to confirm the validity of the model. However, I have the following comments/suggestions, which should help improve the manuscript.

Major
1) While the manuscript uses standard English for most parts, the usage in many places could be improved. For example,
Abstract: “clarify mechanism of disease, and highlight insights into drug discovery.”
Line 71: “multiple deleterious effects on the liver”
Line 86: “Herein we employed a pseudotargeted lipidomic approach to analyze glycerophospholipids by using ultra performance liquid chromatography (UPLC)-hybrid triple quadrupole-linear ion trap (QTRAP)-MS/MS in the time-scheduled multiple reaction monitoring (MRM) mode.”
Line 89: “The present study might provide detailed new insights into the role of scoparone in the regulation of glycerophospholipid metabolism that is of value in understanding the potential treatment of NASH.”
This is to name a few, and there are many other places as such. Thus, I recommend seeking the assistance of a colleague proficient in English and familiar with the subject matter to review your manuscript. Alternatively, you may explore the option of engaging a professional editing service or utilizing AI for assistance.
2) The raw data from the MRM experiments were missing. The intensities and identities of the glycerophospholipids found in the study should be provided.

Minor
3) A space should be added between the text and the open parenthesis of the citations in all places.
4) Lines 77-79: The authors mentioned the studies on the antihyperlipidemic properties of scoparone in CCl4-induced hepatic injury-dependent hyperlipidemia, but did not mention in which animal models the experiments were carried out.
5) Line 125: The full name for H&E should be provided in the heading, as this is when it was first mentioned.
6) Lines 220-221: The full names of PE, LPC, and PC should be spelled out the first time these abbreviations are used.

Experimental design

Major
1) Lines 133-134: I checked the reference cited for the lipid extraction method, and I still think many aspects are still unclear. What is the volume of IPA/MeOH the samples were reconstituted in? What is the amount of internal mix standards added to the samples and when? In general, I think more detail in this method section is greatly needed.
2) Lines 134-135: The last sentence here seems to be so out of place. I believe the method described in the reference already mentioned vortex and centrifugation as well. As other parts of the method were left out, as they were mentioned in the reference already, why was this last sentence written as such?
3) Lines 151-152: The authors mentioned “using the scheduled multiple-reaction monitoring (MRM) mode”. What exactly does this mean, and how exactly was this done? Normally, with MRM, the transitions from the precursor ions to the product ions would need to be specified. What are they that you used? Please put the details into the supplementary information.
4) Line 160-162: I don’t quite follow your calculation formula. Why was V a constant volume at 0.2 mL? Was this the volume of the internal standard mix injected? Also, N = sample volume, which unlike V, you did not specify how much. Wasn’t the sample volume supposed to be the same for all samples?
5) Lines 195-196: The authors mentioned a total of 340 glycerophospholipids detected. What are they? How did you know? How were they identified, considering that only 42 standards were used? I’m a little confused with your method of analysis; What exactly does pseudotargeted analysis mean?
6) Following from the last points, how were the glycerophospholipids identified exactly (especially in lines 220-223)? Please provide the necessary data to support the identification.

Minor
7) Line 116: Can you report the centrifugal speed in g (rcf), rather than in rpm, as rpm for different centrifuges would not be the same?
8) Line 129: I’m not sure if the liver sections should be stained with both hematoxylin and Oil Red O here or with Oil Red O only.
9) Line 143: Do you mean “the column temperature is 55 C”?

Validity of the findings

Major
1) In the abstract, the authors wrote, “Altogether, these results suggested that scoparone may present a novel therapeutic opportunity in the treatment of NASH.” I don’t know if this statement is valid, as in the first sentence, the authors already mentioned that scoparone is the bioactive component of Yinchenhao with known therapeutic effects on NASH. Thus, I don’t think the findings shown in this study provided any “novel” therapeutic opportunity in the treatment of NASH as claimed. This sentence should be revised or more explanations would be needed to support this claim. The same applies to the last sentence of the conclusions (lines 293-294).
2) Lines 207-208: The authors mentioned “14 glycerophospholipids associated with HFD-induced mouse model were screened”. What exactly does this mean? Can you explain why these 14 phosphoglycerolipids when you could detect 340 of them?

---

## Round 0.2 · accepted · Accept

Thank you for the careful attention to all the points raised. I am delighted to recommend acceptance now.

Reviewer 2 ·

Basic reporting

The authors have satisfactorily addressed all my questions in the previous review.

Experimental design

The authors have satisfactorily addressed all my questions in the previous review.

Validity of the findings

The authors have satisfactorily addressed all my questions in the previous review.

Additional comments

The first page of the manuscript seems to have an irrelevant text in a textbox appearing around the title area.